

# Impact of dormancy periods on some physiological and biochemical indices of potato tubers

Hao Liu[1,*], Junhua Li[1,*], Duanrong Zhou[1], Wanhua Cai[1], Muzammal Rehman[2], Youhong Feng[1], Yunxin Kong[1], Xiaopeng Liu[1], Shah Fahad[3] and Gang Deng[1]

[1] School of Agriculture, Yunnan University, Kunming, Yunnan, China
[2] Guangxi Key Laboratory of Agro-environment and Agric-products safety, Key Laboratory of Plant Genetics and Breeding, College of Agriculture, Guangxi University, Nanning, Guangxi, China
[3] Department of Agronomy, Abdul Wali Khan University, Mardan, Khyber Pakhtunkhwa, Pakistan
[*] These authors contributed equally to this work.

Corresponding authors
Shah Fahad, shahfahad@uoswabi.edu.pk
Gang Deng, denggang1986@ynu.edu.cn

## ABSTRACT

**Background.** Storage of potato tubers is an essential stage of the supply chain, from farm to consumer, to efficiently match supply and demand. However, the quality and yield of potatoes are influenced by physiological changes during storage.

**Methods.** This study tested the physiological and biochemical indices in three potato varieties (YunSu 108, YunSu 304 and YunSu 306) during their dormancy periods.

**Results.** Three potato varieties with different dormancy periods were used to follow changes in starch, protein and several enzymes during storage. The starch and sugar content of the long-dormant variety (YunSu 108, LDV) were stable, whereas those of the short-dormant variety (YunSu 306, SDV) were variable. Starch synthase activity in the three varieties was initially high, then decreased; the starch content of LDV was relatively stable, that of the medium-dormant variety (YunSu 304, MDV) increased with storage time and peaked at sprouting, and that of SDV was low but variable. The sucrose synthase activity of LDV was significantly higher ($p < 0.05$) than MDV and SDV in the middle storage period. Two spikes were observed in the invertase activity of SDV, whereas those of MDV and LDV were stable. The reducing sugar content of LDV increased significantly before sprouting, that of MDV slowly decreased and that of SDV dropped sharply. During the whole storage period, pectinase activity in LDV did not change significantly, whereas pectinase in MDV and SDV decreased. The cellulase and protein contents initially increased and then decreased in LDV, and steadily decreased in MDV and SDV.

**Conclusion.** The metabolic indices related to starch and sugar in the LDV were relatively stable during storage, whereas those of the SDV varied greatly. SDV showed increased sucrose, reducing sugars and cellulose; LDV PCA plots clustered in the positive quadrant of PC1 and the negative quadrant of PC2, with increased protein, sucrose synthase and starch; MDV had increased soluble starch synthase.

## INTRODUCTION

Potato (*Solanum tuberosum* L.) is one of the oldest cultivated crops in the world, an annual herbaceous plant from the Solanaceae family and cultivated in South America for more than 8,000 years. Potato is the third-largest food crop in the world after maize and rice, with a total yield of 359 million tons annually in 2019. China is the largest potato producer, with a yield of 18.30 million tons, accounting for about 1/20 of the global total (FAOSTAT, 2021). There are more than 4,500 potato varieties (with different textures, tastes, shapes, colors and sizes) distributed in more than 160 countries and consumed by more than 1 billion people worldwide (*Dourado et al., 2019*).

Potato tubers naturally go into a dormant state during the winter, thereby facilitating storage after harvest and enabling matching of supply with consumer demand. Storage of potatoes during their dormancy period is, therefore, essential to avoid wastage and maintenance of quality during storage, as it is vital to meet consumer expectations. The dormancy period varies greatly from 30 to 150 days, for different varieties or the same variety under different storage conditions. Chemical or physical treatments which delay sprouting can extend storage time, but they are difficult to control and subject to environmental and food safety concerns (*Jakubowski & Krolczyk, 2020*; *Wang, Brandt & Olsen, 2016*). Losses of stored potatoes, caused by dormancy problems, such as the accumulation of toxins, decreased quality and a decrease in dry matter, can be up to 20% (*Santos et al., 2020*; *Sorce et al., 2005*; *Visse-Mansiaux et al., 2021*). Understanding of the physiological processes and regulatory mechanisms of potato dormancy is limited and the current knowledge has not enabled significant improvements to potato storage and reduction of losses.

The development of potato tubers consists of four main stages; tuber initiation, enlargement, dormancy and sprouting to form a new plant (*Fernie & Willmitzer, 2001*). Dormancy has been described as a temporary suspension of visible growth of any plant structure containing a meristem, such as roots and shoots (*Lang et al., 1987*; *Aksenova et al., 2013*). Several factors are responsible for the regulation of tuber dormancy and sprouting, including internal physiological factors and environmental factors (*Rentzsch et al., 2012*; *Sonnewald & Sonnewald, 2014*). Cytokinins; *i.e.,* plant hormones, such as abscisic acid and ethylene are involved in the induction and maintenance of tuber dormancy as well as the inhibition of sprouting, whereas gibberellins, such as gibberellic acid, and cytokinins stimulate sprouting. In addition, changes in endogenous IAA and GA contents in potato tubers may be closely related to the subsequent sprout growth regulation. (*Fernie & Willmitzer, 2001*; *Suttle, 2004*; *Weiner et al., 2010*). Sprouting results in potato tubers becoming soft and shriveled, and production of toxic alkaloids, which renders them inedible; sprouting during storage is a major cause of potato loss (*Sorce et al., 2005*). A standard for the effect of potato sprouting on their quality has been established by the United States Department of Agriculture (USDA, https://www.federalregister.gov/documents/2011/06/02/2011-13485/united-states-standards-for-grades-of-potatoes); quality is significantly affected when buds longer than 0.64 cm (1/4 inch) can be seen on 5% of stored potato tubers.

Dormancy and sprouting of potatoes are regulated by internal and environmental factors. During storage, the environment is relatively stable, so internal physiological factors are dominant. Past research on potato dormancy has identified many substances (important nutrients, such as starch, protein and sugar) and various enzymes, which vary in content during dormancy. However, it is unclear how these substances affect, or are affected by, dormancy of potatoes during storage. Therefore, this study compared the changes in nutrients and enzymes during potato dormancy in three varieties during their dormancy periods, to improve understanding of the relationships between physiological characteristics and potato dormancy. The findings provide an important theoretical reference for potato dormancy research and long-term potato storage in the future.

## MATERIALS & METHODS

### Test materials

The tested potatoes were the main varieties cultivated in Yunnan province, China; namely, YunSu 108(LDV): male parent: S04-827 (from China), female parent: DianSu 6 (from Russia), YunSu 304 (MDV): male parent: yakhant (from Belarus), female parent: 387,136.14 (from China), YunSu 306 (SDV): male parent: ChunSu 3 (from China), female parent: HeZuo 88 (from China). All potatoes were provided by the Grain Crop Institute of Yunnan Academy of Agricultural Sciences (Kunming, China) and cultivated in the potato base of Yunnan Academy of Agricultural Sciences.

### Sample processing

The potato tubers were harvested at 110 days after planting. Tubers of the varieties were selected for absence of damage, disease or decay, similar weight (LDV 210–230, MDV 120–135 and SDV 130–145 g) and similar appearance, then spread in a ventilated, dark place to dry, after these treatments, all the potato tubers went into the endodormancy stage (*Lang et al., 1987*).

For the storage trial, the tubers were placed in a shaded artificial climate box, at 25 °C and sampled every 7 days, with one cm wide square sampler. A 0.5 cm deep core sample was taken at the bud eye. Each variety was sampled until two mm long buds were visible on more than 50% of the bud eyes, which was defined as the start of sprouting/end of dormancy (Fig. S1). This resulted in nine samples from YunSu 108 (LDV) over 56 days, eight from YunSu 304 (MDV) over 49 days and six from YunSu 306 (SDV) over 35 days (Fig. S2). All samples were immediately frozen with liquid nitrogen, ground to powder and then stored in a −80 °C freezer until needed.

### Determination of physiological and biochemical indices
#### Starch metabolism-related indices

All tests were performed using a 0.1 g sample of potato tuber (three biological replicates). The detailed measurements are as follows:

Samples were extracted in 50 mM HEPES-NaOH buffer (pH7.4), homogenized, and centrifuged at 10,000 g for 5 min, The supernatant was used for the experiment. Measurements of the activities of soluble starch synthase activities were made in accordance

with the methods published by *Jiang, Dian & Wu (2003)*, the soluble starch synthases were determined in the supernatant fluids at 340 nm on a spectrophotometer.

Samples were extracted in 0.8 mL distilled water, homogenized, take supernatant, and add DNS solution to solution the amylase. The amylase activities were tested at 540 nm on a spectrophotometer according to the method of *Dziedzoave et al. (2010)*.

Samples were extracted in mL distilled water, homogenized, take supernatant. Take a 50 µL sample from the supernatant and add 250 µL anthrone buffer, after 10 min of 95 °C water bath, the starch content was measured at 620 nm on a spectrophotometer according to the method of *Clegg (1956)*.

### Sugar metabolism-related indices

All tests were performed using a 0.1 g sample of potato tuber (three biological replicates). The detailed measurements are as follows:

Samples were extracted in 50 mM HEPES-NaOH buffer (pH7.4), homogenized, and centrifuged at 8,000 g for 5 min, 210 µL 30% hydrochloric acid and 60 µL 0.1% resorcinol were added into the supernatant, and the mixture was mixed in water bath at 80 °C for 20 min, the sucrose synthetase activities were measured at 620 nm on a spectrophotometer according to the method of *Pressey (1969)*.

Sucrase and reducing sugar content samples were extracted in HEPES-NaOH and 70% ethanol especially, centrifuged at 8,000 g for 10 min, take supernatant, add DNS solution to the supernatant. After that, the sucrase activities and reducing sugar contents were tested at 540 nm on a spectrophotometer according to the methods of *Karley, Ashford & Minto (2005)* and *Lindsay (1973)*.

### Pectinase, cellulase and protein parameters

All tests were performed using a 0.1 g sample of potato tuber (three biological replicates). The detailed measurements are as follows: pectinase, cellulase and protein contents samples were extracted in Gly-NaOH buffer, citric acid buffer, potassium phosphate buffer especially, centrifuged at 8,000 g for 10 min, take supernatant, pectinase and cellulase add DNS solutions, while protein add biuret reagent to the supernatant. After that, the pectinase activities, cellulase activities and protein contents were same tested at 540 nm on a spectrophotometer according to the methods of *Paludo & Kruger (2011)*, *Faria, Kolling & Camassola (2008)* and *Bradford (1976)*.

### Statistical analysis

The triplicate data were subjected to one-way analysis of variance and least significant difference using SPSS 19.0 software (SPSS Inc., Chicago, IL, USA). The significance level was set at $p < 0.05$. Pearson's correlation was used to quantify relationships between different variables. Pearson correlation coefficients and principal component analysis between the potato varieties were calculated using SPSS 19.0 software.

## RESULTS

### Changes in starch metabolism-related indices of potato tubers during storage

For all three varieties, the soluble starch synthase activity was high initially, then decreased gradually until the end of storage (Fig. 1), whereas amylase was relatively stable. The starch content of the three varieties varied differently for the three varieties. The long-dormant variety (LDV, YunSu 108) showed a slight downward trend, medium-dormant variety (MDV, YunSu 304) a moderate upward trend and short-dormant variety (SDV, YunSu 306) an initial marked decrease, followed by a marked increase to a peak at 28 days.

The starch synthase activity in the LDV was highest on day 7 of storage (4.12 U/g), significantly higher than any other sampling day, and decreased gradually to <1.39 U/g after 42 days. Similarly, activity in the MDV was highest (2.68 U/g) on day 7 of storage and decreased to <1.56 U/g after 42 days. The activity in SDV decreased significantly from day 0 to a minimum (0.57 U/g) at day 21, then increased slowly.

The amylase activity of the LDV was maximal (5.04 U/g) at day 0, approximately halved by day 7 and remained essentially stable until a significant increase to 3.16 U/g at day 56. The amylase activity in the MDV was relatively stable, apart from a significant increase to a peak of 3.19 U/g on day 7 of storage. The amylase activity of the SDV was lowest at day 0, increased to a maximum of 3.67 U/g on day 14, significantly higher than other sampling days, then decreased and remained stable. Generally, the three varieties had similar amylase activities.

The starch content of the LDV gradually decreased, apart from a sharp decrease to a minimum of 48.91 mg/g at day 7, followed by a sharp increase to a maximum (113.81 mg/g) on day 14. The starch content of the MDV showed a similar sharp decrease at day 7, but then increased to a maximum on day 49. The starch content of the SDV was highly variable, with a minimum at day 14 and a maximum at day 28, followed by a sharp drop.

### Changes of sugar metabolism-related indices of potato tubers during storage

Sucrose synthase activity in the LDV was relatively stable, except for days 21, 28 and 35, reaching a maximum at day 28 (445.89 U/g) (Fig. 2). The activity in the MDV was stable during the whole storage period. The activity in the SDV was lowest at day 0 of storage, followed by day 35 and was a little higher in between.

The sucrase activity in all three varieties was similar and relatively stable except for the LDV, which was significantly higher than other sampling days at 168.01 U/g on day 7 and the SDV, which spiked at days 14 and 28.

The reducing sugar content of the LDV was stable except for day 49 (391.18 µg/g), which was significantly higher than other sampling days. The content of the MDV gradually decreased, with the highest content on day 7 (368.03 µg/g) and the lowest at day 49, when sprouting started (277.34 µg/g). The content of the SDV was the lowest at day 0 (286.51 µg/g), increased markedly to a maximum on day 7 (469.45 µg/g), significantly higher than all other sampling days, then gradually decreased.

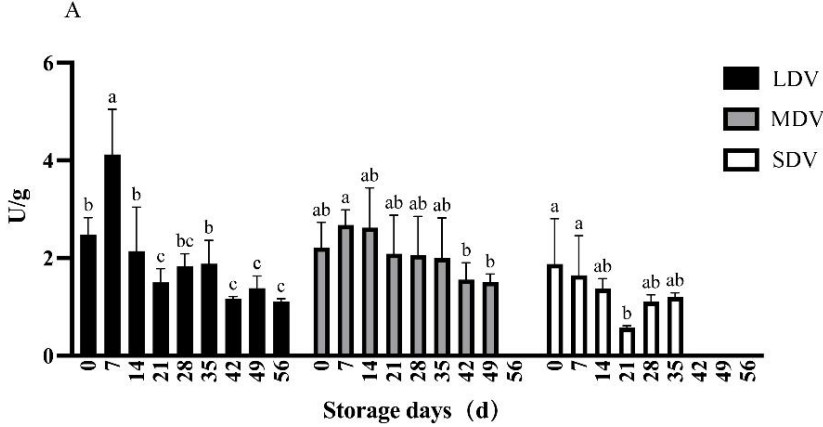

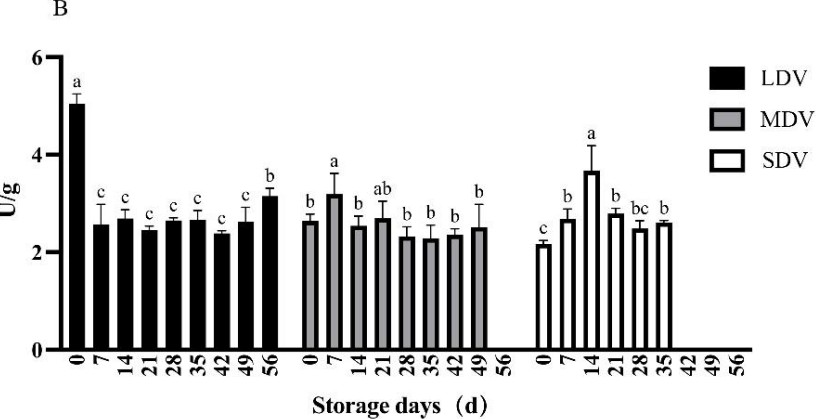

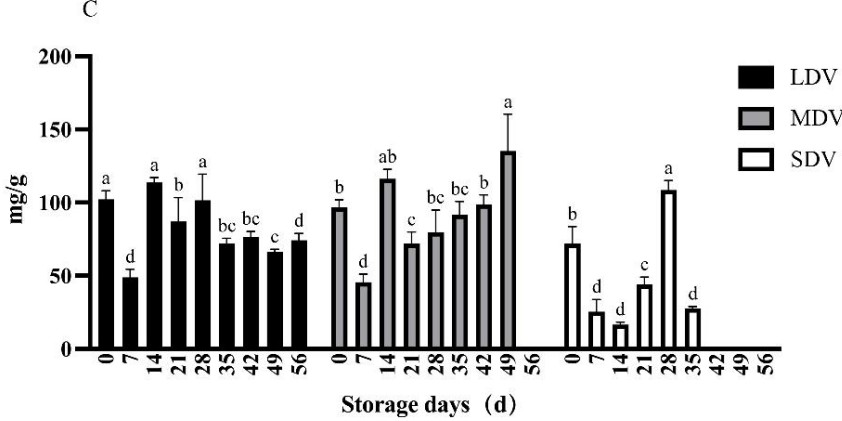

**Figure 1** (A–C) Changes in starch metabolism-related indices (soluble starch synthase and amylase enzyme activities in units (U)/gram, starch content in mg/g) in tubers of three potato varieties during dormancy/storage.

A

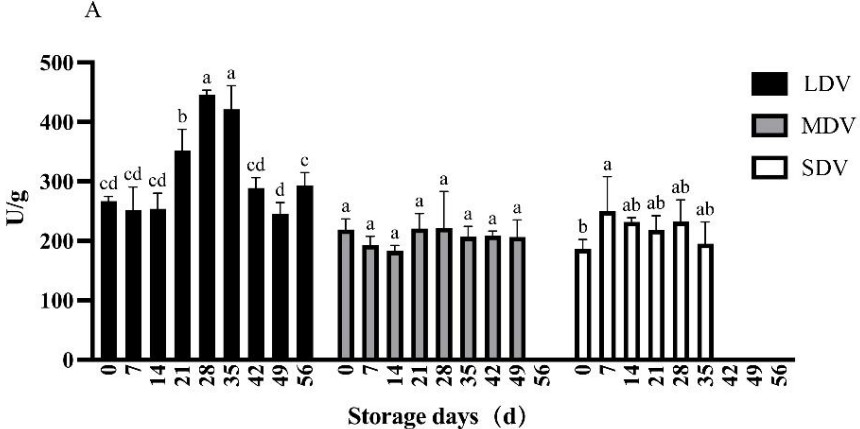

B

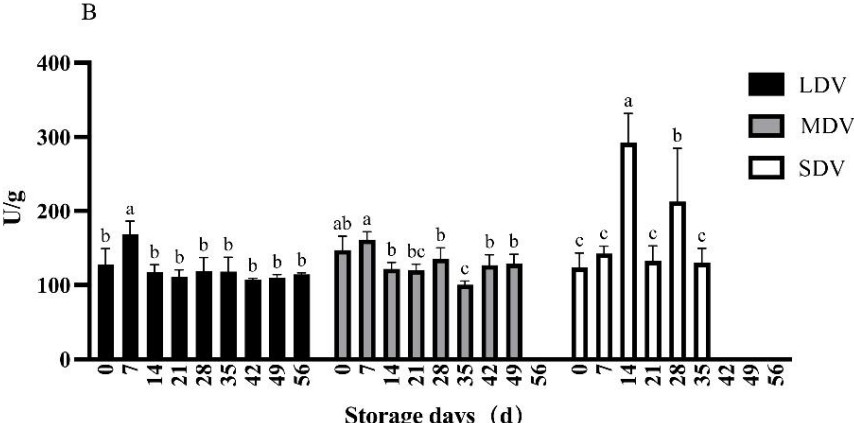

C

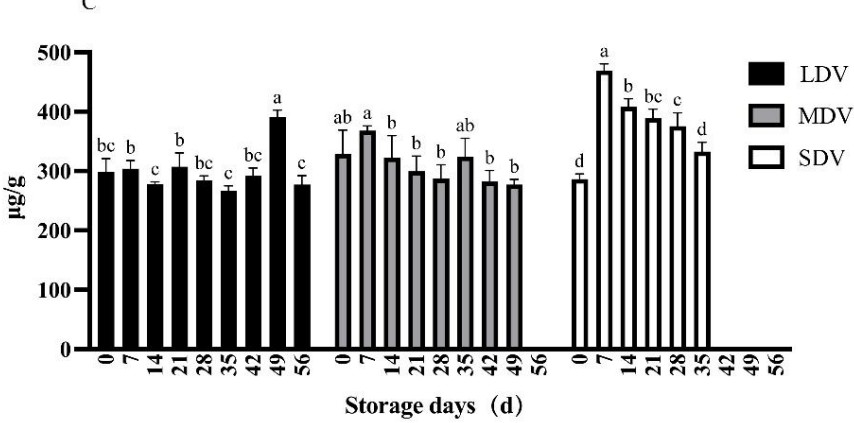

**Figure 2** (A–C) Changes in sugar metabolism-related indices (sucrose synthetase, sucrase and reducing sugar content) in tubers of three potato varieties during dormancy/storage.

## Changes of other metabolic indices of potato tubers during storage

The pectinase activity in the LDV was relatively stable except for non-significant relative increases at 14 and 35 days (Fig. 3). The pectinase activity in the MDV gradually decreased, with a maximum at 14 days, but the differences were not significant. The pectinase activity in the SDV was relatively stable except for a marked decrease at day 7.

The cellulase activity of the LDV showed a general downward trend, with high values on day 7 (220.58 U/g) and day 21 (222.51 U/g), significantly higher than other sampling days, gradually decreasing to a minimum of 142.35 U/g at day 56. The cellulase activity of the MDV was relatively stable with a slight downward trend from a maximum of 248.67 U/g at day 0, significantly higher than at days 14, 35 and 42, and with a small increase at day 49. The cellulase activity in the SDV was relatively stable, with a significant increase to 278.28 U/g at day 35.

The protein content of the LDV was slightly higher in the middle of the storage period, reaching a maximum of 16.92 mg/g at day 42. Conversely, the protein content of the MDV was slightly lower in the middle of the storage period, with a maximum (17.36 mg/g) on day 42, then a marked decrease at day 49. The protein content of SDV gradually, but not significantly, decreased.

## Correlation and principal components analysis

Correlation analysis data for the three potato varieties are presented in Table 1. Correlation showed sucrase and reducing sugar were significantly positively correlated with cellulase, sucrose synthetase was significantly positively correlated with protein and amylase and reducing sugar were significantly positively correlated with sucrase. Soluble starch synthase and starch were significantly negatively correlated with cellulase, starch was significantly negatively correlated with sucrose synthetase, and starch was significantly negatively correlated with sucrose. YunSu 306 (SDV) had increased sucrose, reducing sugars and cellulose. YunSu108 (LDV) PCA plots clustered in the positive quadrant of PC1 and the negative quadrant of PC2, with increased sucrose synthase and starch (Fig. 4). YunSu 304 (MDV) plots had increased soluble starch synthase. A strong negative correlation between the contents of starch and reducing sugar was observed. A weak negative correlation as observed between amylase and soluble starch synthase.
Add your results here.

## DISCUSSION

Generally, dormancy is classified into three types; *i.e.,* relative dormancy, physiological dormancy and environmental dormancy, each with three stages, initiation, maintenance and termination/sprouting (*Gong et al., 2021*; *Vreugdenhil & Bradshaw, 2007*). There are no obvious physiological markers of the different periods, so it is difficult to distinguish them. Normally, relative dormancy and environmental dormancy are mutually reversible under external factors, while physiological dormancy is less likely to be reversed (*Vreugdenhil & Bradshaw, 2007*). Physiologically, the dormancy period of potato tubers starts from the expansion of the tip of the stolon to the start of sprout growth from the buds. However, it is generally accepted that the dormancy of potato tubers starts from the harvest of the

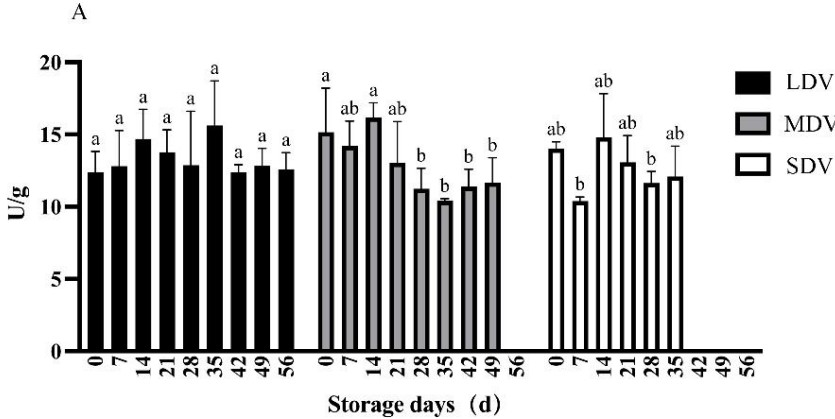

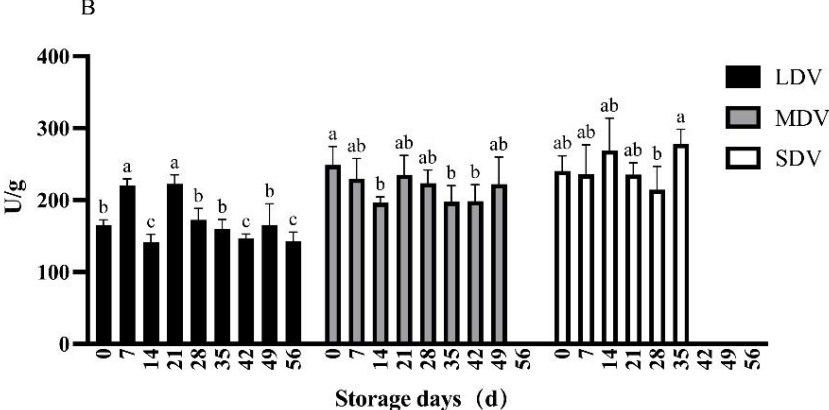

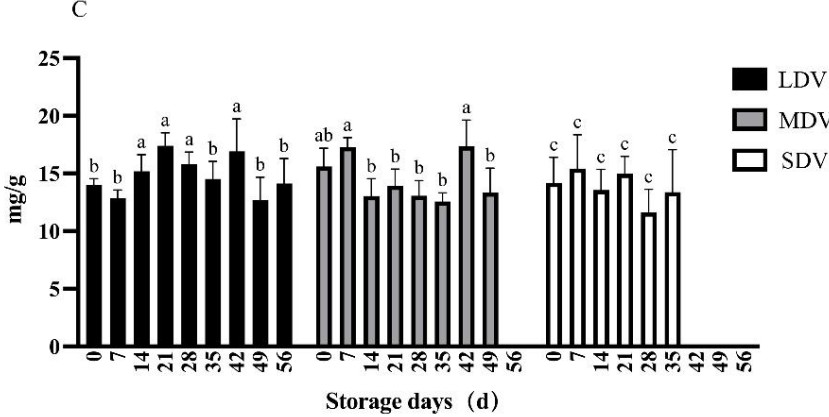

**Figure 3** (A–C) Changes of other metabolism-related indices (pectinase and cellulase activities, and protein content) in tubers of three potato varieties during dormancy/storage.

**Table 1 Correlation among starch, sugar and other metabolic indices of three potato varieties during dormancy/storage ($r^2$).**

|  | PT | SS | CL | SS | SR | AL | PT | ST | RS |
|---|---|---|---|---|---|---|---|---|---|
| PT |  |  |  |  |  |  |  |  |  |
| SS | 0.109 |  |  |  |  |  |  |  |  |
| CL | −0.077 | −0.023 |  |  |  |  |  |  |  |
| SS | 0.04 | −0.046 | −0.415** |  |  |  |  |  |  |
| SR | 0.06 | −0.03 | 0.470** | −0.171 |  |  |  |  |  |
| AL | 0.031 | 0.055 | −0.092 | 0.053 | 0.243* |  |  |  |  |
| PT | 0.05 | −0.059 | −0.003 | 0.242* | −0.18 | 0.015 |  |  |  |
| ST | −0.025 | 0.111 | −0.420** | 0.08 | −0.353** | −0.114 | −0.02 |  |  |
| RS | −0.129 | −0.2 | 0.374** | −0.255* | 0.440** | 0.115 | −0.091 | −0.544** |  |

**Notes.**

PT, pectinase; SSS, soluble starch synthase; CL, cellulase; SS, sucrose synthetase; SR, sucrase; AL, amylase; PT, protein; ST, starch; RS, reducing sugar.

*Significant $r^2$ values at $p < 0.05$.
**Significant $r^2$ values at $p < 0.01$.

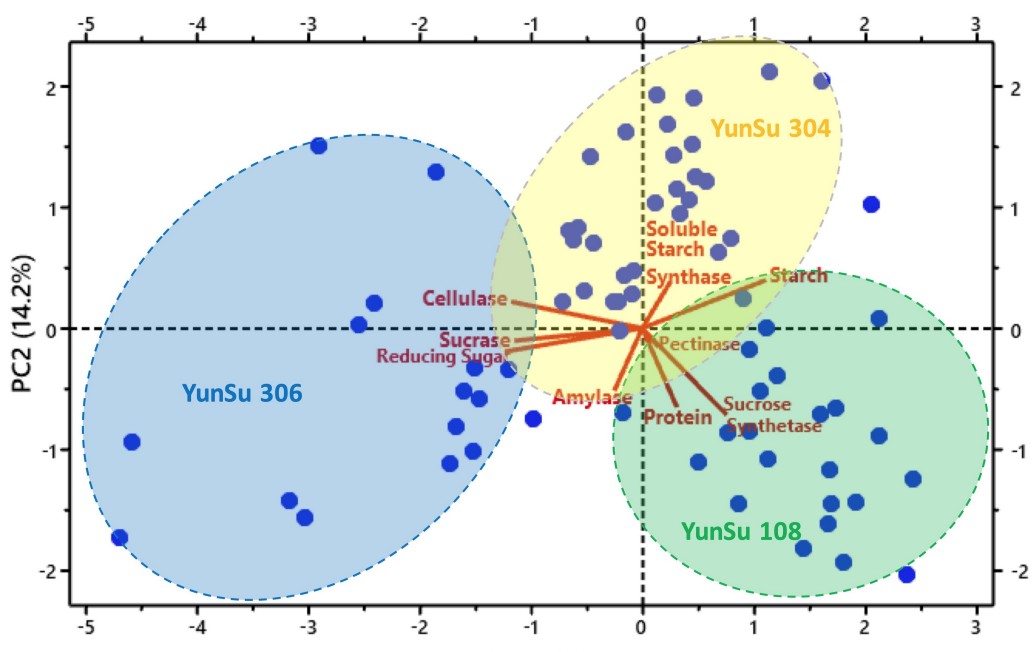

**Figure 4** Principal component analysis (PCA) plot showing three varieties of potatoes and associated variables' clusters.

tubers to the sprouting of their top bud. During dormancy the tissues maintain some physiological and metabolic functions. The tubers can only sprout and grow when the dormancy ends naturally, or is ended artificially by a physical or chemical treatment (*Reust & Gugerli, 1984*).

Starch is the main carbon/energy storage substance and changes in starch content affect the quality of potatoes (*Gong et al., 2021*) The starch-related indices of the three potato varieties were compared during storage; the starch synthase and amylase activities,

and starch content of the LDV were relatively stable, whereas those of the SDV varied widely. The starch synthase activity of the three varieties was relatively high at the start of storage, then decreased until sprouting, reflecting a decrease in starch biosynthesis, in agreement with a reported decrease in starch content of potatoes at different storage temperatures (*Wang et al., 2020*).The starch synthase activity of the SDV trended upwards during storage, whereas its amylase activity trended downwards; the starch content was initially at a medium level, decreased to a low level, then rapidly increased to a maximum just before sprouting. This suggests that the metabolism of carbohydrates was affected (*Zhao et al., 2004*), thereby directly inducing sprouting.

Starch degradation generally occurs in potatoes during storage and it is converted into sugars or other substances; the decreased starch content causes the tubers to shrink and become wrinkled (*Zhao et al., 2004*). Increased reducing sugar content, especially at low temperatures, lowers the quality of stored potatoes (*Jakubowski & Krolczyk, 2020*). As for starch metabolism, the sugar metabolism-related indices of the SDV varied more widely than those of the other two varieties. The LDV had a significant increase in sucrose synthase activity in the middle storage period (from day 21 to day 35), followed by a sharp decrease, while its invertase activity remained stable, indicating that the rate of starch biosynthesis is high in the middle period of dormancy. In addition, there were two spikes in invertase activity in the SDV during storage, whereas that of the other two varieties remained stable; the spikes may be associated with the end of dormancy and the initiation of sprouting in the SDV. These findings indicate that starch saccharification in potatoes may be attributed to increased enzyme activity, mainly sucrase, during storage (*Farrar, Pollock & Gallagher, 2000*; *Shu et al., 2017*), and that starch saccharification may be related to the end of dormancy and the initiation of sprouting. The reducing sugar content of in the MDV decreased slowly during the storage period, whereas that in the SDV increased sharply during the first week, then decreased markedly. The LDV was the only variety which had a significant increase in reducing sugar content before sprouting, then a decrease to the initial content after sprouting, This finding is different from a previous report that the total sugar content of potatoes increased before tuber sprouting (*Abbasi et al., 2015*). The LDV may also be affected by other metabolites and enzymes, resulting in steady interconversion between reducing sugar and starch, prolonging dormancy (*Campbell, Suttle & Sell, 1996*; *Zhang et al., 2014*), which is consistent with the observed changes in saccharification-related enzymes in the LDV.

The changes in pectinase activity were significantly different among the three varieties. Activity in the LDV was relatively stable during the whole storage period, that in the MDV markedly decreased and that in the SDV gradually decreased. All three varieties had decreased pectinase activity before and during sprouting, suggesting that pectinase has a regulatory function in maintaining dormancy, which may be related to its function of maintaining cell wall integrity (*Ishibashi et al., 2017*). The cellulase activity in the three varieties was relatively stable but characterized by different trends. In the LDV, cellulase activity initially increased, then decreased, whereas that in the MDV and SDV gradually increased towards sprouting. Cellulase may also help regulate the maintenance of dormancy, as cellulase catalyzes the formation of glucose from cellulose, thereby helping maintain

reducing sugar homeostasis (*Withers et al., 1990*). The protein content of the LDV initially increased, then decreased, whereas that of the other two varieties decreased gradually; increased protein content in the middle storage period would contribute to maintenance of dormancy. This finding is different from a previous report that the protein content of potatoes does not change significantly during dormancy (*Gong, Zhao & Yang, 2004*).

## CONCLUSIONS

The metabolic indices related to starch and sugar in the LDV were relatively stable during storage, whereas those of the SDV varied greatly. The most important findings were as follows: (1) starch content increased before sprouting in LDV. (2) The fluctuation of starch synthase activity during storage may lead to the breaking of dormancy. (3) The decrease in reducing sugar content during storage was mainly in the early dormancy period. (4) The decrease in pectin and protein content during storage was mainly in the early dormancy period. (5) YunSu 306 (SDV) had increases in sucrose, reducing sugars and cellulose; YunSu108 (LDV) had increases in protein, sucrose synthase and starch; YunSu 304 (MDV) had increased soluble starch synthase. Future research should investigate how modification of storage conditions could be used to control the duration of potato dormancy, based on changes in these key metabolic indices.

### Funding

This research was supported by a grant from the National Natural Science Foundation of China (32160100), the Yunnan Provincial Joint Fund for Local Colleges and Universities (202001BA070001-129), and the Yunnan Applied Basic Research Projects (202201AT070150) and Xingdian Talent Support Plan. The funders had no role in study design, data collection and analysis, decision to publish, or preparation of the manuscript.

### Grant Disclosures

The following grant information was disclosed by the authors:
National Natural Science Foundation of China: 32160100.
Yunnan Provincial Joint Fund for Local Colleges and Universities: 202001BA070001-129.
Yunnan Applied Basic Research Projects: 202201AT070150.
Xingdian Talent Support Plan.

### Competing Interests

The authors declare there are no competing interests.

### Author Contributions

- Hao Liu conceived and designed the experiments, performed the experiments, analyzed the data, prepared figures and/or tables, authored or reviewed drafts of the article, and approved the final draft.

- Junhua Li conceived and designed the experiments, performed the experiments, analyzed the data, prepared figures and/or tables, authored or reviewed drafts of the article, and approved the final draft.
- Duanrong Zhou conceived and designed the experiments, performed the experiments, analyzed the data, prepared figures and/or tables, authored or reviewed drafts of the article, and approved the final draft.
- Wanhua Cai conceived and designed the experiments, performed the experiments, analyzed the data, prepared figures and/or tables, authored or reviewed drafts of the article, and approved the final draft.
- Muzammal Rehman conceived and designed the experiments, performed the experiments, analyzed the data, prepared figures and/or tables, authored or reviewed drafts of the article, and approved the final draft.
- Youhong Feng conceived and designed the experiments, performed the experiments, analyzed the data, prepared figures and/or tables, authored or reviewed drafts of the article, and approved the final draft.
- Yunxin Kong conceived and designed the experiments, performed the experiments, analyzed the data, prepared figures and/or tables, authored or reviewed drafts of the article, and approved the final draft.
- Xiaopeng Liu conceived and designed the experiments, performed the experiments, analyzed the data, prepared figures and/or tables, authored or reviewed drafts of the article, and approved the final draft.
- Shah Fahad conceived and designed the experiments, performed the experiments, analyzed the data, prepared figures and/or tables, authored or reviewed drafts of the article, and approved the final draft.
- Gang Deng conceived and designed the experiments, performed the experiments, analyzed the data, prepared figures and/or tables, authored or reviewed drafts of the article, and approved the final draft.

## Data Availability

The raw measurements are available in the Supplemental Files.

## Supplemental Information

Supplemental information for this article can be found online at http://dx.doi.org/10.7717/peerj.15923#supplemental-information.

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
