# Peer review of "Impact of dormancy periods on some physiological and biochemical indices of potato tubers"

_PeerJ, doi:10.7717/peerj.15923_

## Round 0.1 · original submission · Major Revisions

Revise the manuscript carefully.

·

Basic reporting

The research explores the complex biochemical interrelationships in three types of potato varieties and tuber dormancy. The methods used to meet the objective set by the researchers include the analysis of some parameters as indicators of starch metabolism and tuber sugars.
The results, however, are limited to the varietal characteristics of the samples; they are local varieties that are not known outside China. Also, the sample size seems to be insufficient to draw more generalized conclusions about biochemical indices and tuber dormancy. What does seem to be very accurate is the linkage between varietal types (long, medium and short dormancy) and some of the indicators indicated.
Regarding principal component analysis, the values of PC1 and PC2 are relatively small (27.9% and 14.2%, respectively) to explain the degree of association between biochemical indicators and varietal characteristics. However, the analysis and presentation of the results are adequate.

Experimental design

The research mentions the tuber sampling process and the type of statistical analysis (one-way ANOVA) for the three varietal samples. A completely randomized design is assumed but is not mentioned in the methodological part.

Validity of the findings

Within the limitations already pointed out in the basic reporting, the results obtained are considered valid for the varietal samples chosen, but it is not possible to make inferences with certainty for other potato genotypes.

Additional comments

According to the results obtained from the research, the reviewer considers that the manuscript is a valuable contribution to research in the storage and conservation of potato tubers by finding some biochemical indicators that are used in the industry to measure the quality of the tuber for processing and for direct consumption.
There is an error in table 1, regarding the correlation coefficient which is symbolized in the table footnote as r2 when it should be only r.

Reviewer 2 ·

Basic reporting

line 36: use of the term "germination" is not usually used when describing potato tuber growth. The term sprouting is more common.

line 71-73: definition of dormancy could be clearer. See Lang et al. 1987

line 77-79: hormonal control of tuber sprouting see the work of Suttle

line 263: "(1) starch content increased before sprouting is related to long dormancy period" is unclear. can you be clearer on what is meant by "long dormancy period". Do the authors mean the potato variety with long dormancy?

Experimental design

line 120: The authors refer to wang 2015 for the methods. The methods should be repeated in this manuscript as readers may not have access to Wang 2015. As a reviewer I do not and cannot, at this time, adequately review the manuscript.

Validity of the findings

manuscript can not be reviewed as written

---

## Round 0.2 · Major Revisions

Address all the comments carefully to avoid any further delay.

Reviewer 2 ·

Basic reporting

The manuscript has improved with respect to word usage and clarity.

Experimental design

There are some major issues with the experimental approach, which make the review of the manuscript problematic. It was suggested that the authors clearly define the stage of dormancy they are examining; endo-dormant, para-dormant, or eco-dormant as defined by Lang et al. In response to the suggestion the authors added a reference for Lang but did not describe the stage of dormancy examined in their experimental approach.

A more serious issue is the methods section, which is severely lacking. It was suggested that they add detail to the methods because the methods outlined were defined by a single reference that was not available to the reviewer. In response the authors added a series of references to define the experimental approach. However, using a series of references does not allow a reader to know the type of tissue examined, the amount of tissue examined, the equipment used for the assays. In other words, it is not possible for a reader, or reviewer, to ascertain the quality of the experimental approach. Thus, the manuscript cannot be adequately reviewed as written. It is requested that the authors include accurate and clear methods.

Validity of the findings

A written the manuscript cannot be adequately reviewed. It is requested that the authors include accurate and clear methods.

---

## Round 0.3 · accepted · Accept

The coments are satisfactorily addressed.

Reviewer 2 ·

Basic reporting

The manuscript has been much improved since the last review.

Experimental design

The experiments outlined are well described and clear. The addition of detailed methods greatly improved the manuscript. The experiment outlined, and the data collected is appropriately, conducted and analyzed.

Validity of the findings

The authors describe changes in starch and sucrose metabolism in early, medium, and late dormancy potato varieties. The data does demonstrates a difference between these varieties. The authors are making the suggestions that there is a relationship between the short, medium, and long dormancy behavior but the changes in starch and sugar metabolism might indicative of dormancy status and not specifically variety dependent. It wold be interesting to see the approach applied to more varieties. The authors present an interesting paper that open up additional questions pertaining to dormancy status and starch and sucrose metabolism.